# Towards Cross-modal Backward-compatible Representation Learning for Vision-Language Models

## Abstract

Modern retrieval systems often struggle with upgrading to new and more powerful models due to the incompatibility of embeddings between the old and new models. This necessitates a costly process known as backfilling, which involves re-computing the embeddings for a large number of data samples. In vision, Backward-compatible Training (BT) has been proposed to ensure that the new model aligns with the old model's embeddings. This paper extends the concept of vision-only BT to the field of cross-modal retrieval, marking the first attempt to address Cross-modal BT (XBT). Our goal is to achieve backward-compatibility between Vision-Language Pretraining (VLP) models, such as CLIP, for the cross-modal retrieval task. To address XBT challenges, we propose an efficient solution: a projection module that maps the new model's embeddings to those of the old model. This module, pretrained solely with text data, significantly reduces the number of image-text pairs required for XBT learning, and, once it is pretrained, it avoids using the old model during training. Furthermore, we utilize parameter-efficient training strategies that improve efficiency and preserve the off-the-shelf new model's knowledge by avoiding any modifications. Experimental results on cross-modal retrieval datasets demonstrate the effectiveness of XBT and its potential to enable backfill-free upgrades when a new VLP model emerges.

## 1 Introduction

As the volume and variety of data grow exponentially in our multimedia-rich era, developing and maintaining efficient multi-modal retrieval systems becomes increasingly challenging. These systems, which provide data samples relevant to a user's query, must handle diverse data types, from text and images to audio and video. This growth puts a premium on the scalability and performance of these systems, necessitating continuous advancements in algorithms and technology. Embedding-based deep models for retrieval have emerged as a key solution, transforming high-dimensional data into a lower-dimensional dense embedding space Rehman et al. (2012); Zhou et al. (2017); Wan et al. (2014); Jang & Cho (2020); Hoe et al. (2021); Jang et al. (2022). These models capture the semantic meanings of data samples, enabling the quantification of similarities for retrieval.

However, it is important to note that the embedding spaces generated by different deep models are not inherently compatible with each other. This incompatibility restricts the reuse of an existing gallery that has been embedded with an older model when a new, better-performing model is introduced. Consequently, this necessitates "*backfilling*," a process where the entire gallery must be rebuilt using the embeddings from the new model. Such a requirement is resource-intensive and time-consuming, posing a significant challenge when building retrieval systems.

Backward-compatible Training (BT) Shen et al. (2020); Zhang et al. (2022a); Hu et al. (2022); Wang et al. (2020); Zhang et al. (2022b) has been developed to tackle this issue, specifically focusing on image retrieval systems. The main objective of BT is to train a new model from scratch in a manner that ensures its compatibility with an old model that was used to create the existing gallery. A successful BT model must consequently demonstrate that retrieving from the old gallery using a query embedded with the new model outperforms that using an embedding from the old model. This enhancement is crucial for justifying the application of BT.

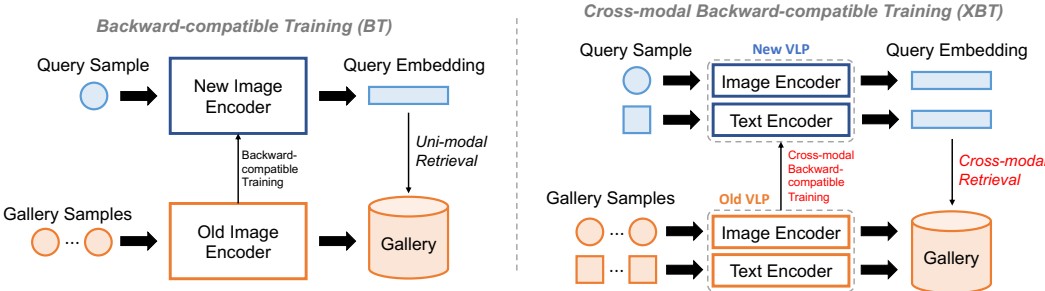

Figure 1: A conceptual visualization of Backward-compatible Training (BT, left) and its extension, the Cross(X)-modal version (XBT, right). Circles and squares denote data samples of images and text, respectively. XBT uses Vision-Language Pretraining (VLP) models as baselines to achieve cross-modal, backward-compatible representation learning, allowing the new, improved model to be compatible with the fixed old model.

Taking the BT problem a step further, we propose a new challenging task called *Cross(X)-modal Backward-compatible Training* (XBT) as shown in Figure 1. Our focus here is on applying BT principles within the realm of cross-modal retrieval, specifically exploring the interaction and compatibility between image and text embeddings of different Vision-Language Pretraining (VLP) models Radford et al. (2021); Li et al. (2022b); Jia et al. (2021); Li et al. (2022a); Singh et al. (2022).

To achieve XBT, just like the BT problem, we need to resolve the incompatibility between the embeddings of the old, inferior model and the new, improved model. If we follow most prior practices of training from scratch with an additional compatibility loss term, a substantial number of supervised image-text pairs would be needed. The ideal quantity would be in the hundreds of millions, approximating the amount utilized in the establishment of VLP models. However, accessing the original training samples is often impossible, and training on that scale is prohibitively expensive and impractical. We propose an efficient solution instead: a text-only *pretrained* projection module, $\phi$, to align a given *pretrained* new model's embeddings with those of the old model.

Our focus is on *using only text data* to estimate the entire distribution of embeddings in the VLP space. Specifically, we train $\phi$ to learn the similarity between old and new text embeddings in a contrastive manner. By increasing the number of text samples, which is simpler than preparing image-text pairs, $\phi$ can approximate the oracle projection between the intra-modal distributions of both old and new embeddings for texts. Assuming that the intra-modal distribution of texts in the VLP embedding space can *mirror that of images*, we can simply apply the same $\phi$ to the new image embeddings to synthesize the corresponding old image embeddings.

With these generated synthetic old image and text embeddings, which can be considered as aligned with the new embeddings, we aim to fine-tune the new model. In this process, far fewer supervised image-text pairs are required than in the original training dataset of the VLP. This also allows us to avoid using old model parameters during training, thereby enhancing training efficiency. In addition, we incorporate parameter-efficient fine-tuning schemes Lester et al. (2021); Kim et al. (2021); Hu et al. (2021) into the new model, which add small trainable parameters and do not harm any of the new model's original parameters during training. This not only accelerates the new model's fine-tuning process but also allows for an easy reversion to the original new-to-new retrieval by simply removing the additional parameters.

We demonstrate our approach on various cross-modal benchmarks, highlighting the effectiveness of XBT in cross-modal retrieval protocols. In response to the rapid advancement of VLP models, XBT offers an environmentally friendly solution by enabling backfill-free systems.

Our contributions can be summarized as:

- The Cross-modal Backward-compatible Training (XBT) concept is introduced for the first time to solve the backfilling problem that stems from real-world cross-modal retrieval systems.

- A novel XBT solution is proposed, which uses a text-only pretrained projection module, $\phi$, to efficiently align the embeddings of new and old models using only text samples.
- The proposal is demonstrated on various datasets and protocols, showcasing XBT's effectiveness in building backfill-free cross-modal retrieval systems.

## 2 RELATED WORKS

**Backward-compatible Training.** The concept of Backward-compatible Training (BT) was first introduced in the study (Shen et al. (2020)). This approach influences a new model with the learned classifier of the old model, thereby achieving backward compatibility between the old and new models. However, BT can degrade the original representational performance of the new model. To overcome this, Meng et al. (2021) proposed aligning class-wise centers presented by the old and new models. Another approach to achieve backward compatibility is to use an additional projection to map the old embedding into the new embedding space by adding a lightweight transformation, as suggested in Hu et al. (2022); Wang et al. (2020). The work of Ramanujan et al. (2022) further adds an auxiliary feature in preparation for future updates, while Zhou et al. (2022) uses additional dimensionality in the embedding to obtain compatibility. An online strategy that backfills the gallery on the fly is explored in Zhang et al. (2022a), and Yan et al. (2021) addresses the model regression problem. Despite this progress, the scenario considering cross-modal retrieval between image and text, which has many real-world applications, remains unexplored. Our study on XBT in this paper aims to fill this gap.

**Vision-Language Continual and Transfer Learning** The fields of continual learning Aljundi et al. (2017); Chaudhry et al. (2019); De Lange et al. (2021) and transfer learning Lu et al. (2015); Zhuang et al. (2020) bear similarities to backward-compatible representation learning, as all aim to update an existing model to boost performance. In the realm of multi-modal representation learning for cross-modal retrieval, continual learning approaches like Wang et al. (2021) propose methods to prevent catastrophic forgetting across different modalities. Transfer learning approaches like Zhen et al. (2020) suggest strategies for transferring knowledge from previously labeled categories (source modality) to new, unlabeled categories (target modality). However, our proposed XBT stands apart from these methods as it specifically tackles the challenge of maintaining backward compatibility between old and new models. This unique attribute makes XBT ideally suited for backfill-free retrieval scenarios, where the embeddings of the old model remain unchanged, yet we can still leverage the enhanced performance of the new model.

## 3 METHODOLOGY

Our goal is to construct a backfill-free, embedding-based, cross-modal retrieval system using VLP models, which are configured with an image encoder $E^I$ and a text encoder $E^T$. When a new better performing VLP model $\{E^I_{new}, E^T_{new}\}$ emerges, we aim to ensure its compatibility with the old model $\{E^I_{old}, E^T_{old}\}$ that was used to construct the gallery. To achieve this, we introduce Cross(X)-modal Backward-compatible Training (XBT). Given an image $x$ and text caption $t$, XBT enables retrieval between a new image embedding $v_{new} = E^I_{new}(x)$ and the text embeddings ($w_{old}$), as well as between a new text embedding $w_{new} = E^T_{new}(t)$ and the image embeddings ($v_{old}$). We denote backward compatible embeddings as $\bar{v}$ and $\bar{w}$. All embeddings we utilize in this work are $l2$-normalized.

### 3.1 CRITERION FOR CROSS-MODAL BACKWARD COMPATIBILITY

Following the definition of backward compatibility in BT work Shen et al. (2020), we can construct strict constraints that ensure cross-modal backward compatibility as:

$$d(w_{new_i}, v_{old_j}) \leq d(w_{old_i}, v_{old_j}), \forall y_i = y_j, d(w_{new_i}, v_{old_j}) \geq d(w_{old_i}, v_{old_j}), \forall y_i \neq y_j,$$
$$d(v_{new_i}, w_{old_j}) \leq d(v_{old_i}, w_{old_j}), \forall y_i = y_j, d(v_{new_i}, w_{old_j}) \geq d(v_{old_i}, w_{old_j}), \forall y_i \neq y_j, \quad (1)$$

where $y_i$ and $y_j$ represent whether an image and text are paired ($y_i = y_j$) or not ($y_i \neq y_j$). The term $d(\cdot, \cdot)$ represents a distance metric in the embedding space, and we choose cosine distance as the

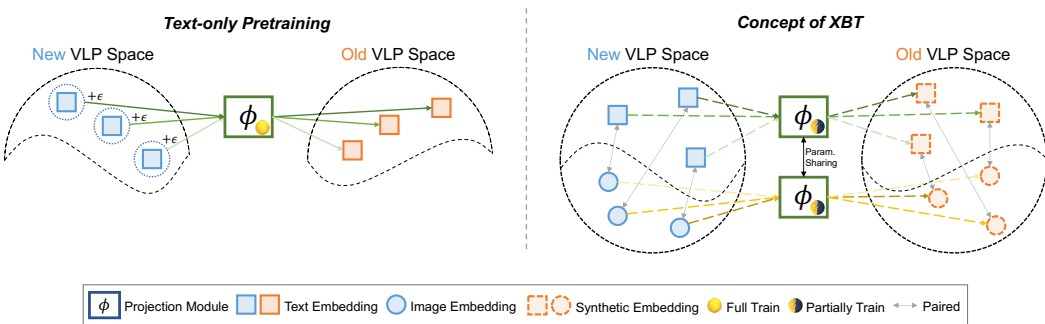

Figure 2: An illustration of the text-only pretraining of $\phi$ (left), and XBT with $\phi$ (right). Using only text samples, $\phi$ is trained to approximate distribution of old text embeddings from that of new ones. During training, noise $\epsilon$ is injected to input of $\phi$. After pretraining, the same $\phi$ is used to generate both of synthetic old image and text embeddings from the new VLP embeddings and train to learn cross-modal backward-compatible representation.

baseline. The constraints in Eqn. 1 formally express the requirement that the new embedding must perform at least as well as the old embedding in terms of correctly matching image-text pairs.

However, exhaustively satisfying these constraints is intractable due to the potential for the old embeddings to outperform the new embeddings in certain retrieval cases. As a result, we relax the criteria by defining an alternative evaluation metric as:

$$\begin{aligned}
\mathcal{M}(w_{new}, v_{old}; Q, G) > \mathcal{M}(w_{old}, v_{old}; Q, G), \\
\mathcal{M}(v_{new}, w_{old}; Q, G) > \mathcal{M}(v_{old}, w_{old}; Q, G),
\end{aligned} \tag{2}$$

where a metric, such as recall, $\mathcal{M}(q, g; Q, G)$ takes query embedding $q$ and gallery embedding $g$ over query set $Q$ and gallery set $G$. This criterion suggests that the general performance is enhanced when the query embedding from the XBT-trained new model is used to perform retrieval with the gallery of the old model, compared to the performance of the old model alone. In essence, fulfilling Eqn. 2 indicates that the new model has achieved backward compatibility and can feasibly update without backfilling gallery.

Additionally, it is important to highlight that VLP models function as zero-shot learners. This leads us to define the XBT problem differently from classic BT, which trains separate models for specific image domain retrieval tasks such as ImageNet Russakovsky et al. (2015) or VGGFace2 Cao et al. (2018). In contrast, XBT tackles a more challenging task, aiming to bridge a new VLP model with a frozen old VLP model while retaining the new model's original zero shot capability. In this paper, we therefore assess the performance of XBT using retrieval and classification benchmarks in a zero-shot manner.

## 3.2 TEXT-ONLY PRETRAINING

The vast and diverse image-text pairs used to build VLP models significantly enhance their ability to connect semantically similar visual content and natural language Chen et al. (2023). However, this creates a complex embedding distribution that is difficult to predict and understand, thereby complicating the XBT process. A straightforward solution is to use a large number of supervised image-text pairs, similar to the approach used when building VLP models from scratch, to estimate the entire distribution of image and text embeddings during XBT.

However, acquiring a sufficient number of accurate supervised pairs is extremely costly. To be far more efficient, we employ a small sized projection module, $\phi$, and train it exclusively with text samples, as shown in Figure 2. We hypothesize that the distribution of text embeddings in VLP models, which is determined by their semantic similarity, is similarly mirrored in the distribution of their corresponding matched images. With this in mind, we aim to train $\phi$ to cover the broad spectrum of the text embedding distribution between the new and old VLP models, making embeddings from the same text sample similar and others dissimilar in a contrastive way:

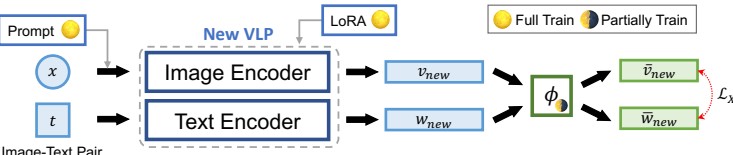

Figure 3: Our proposed learning process to achieve XBT. Notably, the old VLP model's encoders are not required in this stage, enhancing efficiency in training.

$$\mathcal{L}_{pre} = \mathbb{E}_{t \sim D_T}[\mathcal{L}_c(\phi(w_{new} + \epsilon), w_{old}; \tau_{pre})], \tag{3}$$

where $t$ denotes text data sampled from text corpus $D_T = \{t_i\}_{i=1}^{N_{D_T}}$ of $N_{D_T}$ text samples, and $\mathcal{L}_c(\cdot, \cdot)$ is a standard batch-wise contrastive loss as defined in CLIP Radford et al. (2021) of temperature $\tau_{pre}$.

To prevent overfitting towards the text domain, which could widen the discrepancy between the image and text embedding representations, we inject noise $\epsilon$ sampled from a Gaussian distribution ($\epsilon \sim N(0, \sigma^2)$), inspired by the approaches Nukrai et al. (2022); Gu et al. (2023). Notably, $\phi$ is designed to take the new VLP model's embeddings, which means that $w_{new} + \epsilon$ is also $l2$-normalized.

During this stage, only $\phi$ is updated, while all encoders remain fixed. By expanding the scale of $D_T$, we can effectively replicate the complete text embedding spaces of both new and old VLP models. Additionally, $\phi$ can be applied to new image embeddings to generate synthetic old image embeddings that closely approximate the actual old ones.

### 3.3 CROSS-MODAL BACKWARD-COMPATIBLE TRAINING

**Training Loss.** The objective of XBT is to ensure cross-modal compatibility, specifically between $w_{new}$ and $v_{old}$, as well as between $v_{new}$ and $w_{old}$. For a given dataset $D = \{x_i, t_i\}_{i=1}^{N_D}$ of $N_D$ supervised image-text pairs, we aim to train the new VLP model encoders, $E_{new}^I$ and $E_{new}^T$. Note that in our base setting, the text corpus is significantly larger than the supervised dataset, i.e., $N_{D_T} \gg N_D$.

However, the dimensionality of new and old VLP embeddings may differ, and even if they are the same, they are not be directly compatible. To address this, we apply the pretrained $\phi$ to ensure that the new embeddings match the dimension of the old ones and project into the compatible embedding space, as shown in Figure 2. This can be represented as:

$$\phi(v_{new}) = \bar{v}_{new}, \phi(w_{new}) = \bar{w}_{new}, \tag{4}$$

where $\bar{v}_{new}$ and $\bar{w}_{new}$ denote synthetic old embeddings and $\{\bar{v}_{new}, \bar{w}_{new}, v_{old}, w_{old}\} \subset \mathbb{R}^K$, with $K$ denoting the dimensionality of the old VLP model's embedding.

We then apply XBT loss $\mathcal{L}_X$ as follows:

$$\mathcal{L}_X = \mathbb{E}_{x,t \sim D}[\mathcal{L}_c(\bar{v}_{new}, \bar{w}_{new}; \tau_X)]. \tag{5}$$

Here, $\mathcal{L}_c(\cdot, \cdot)$ is the same contrastive loss used in Eqn. 3, with temperature $\tau_X$. As shown in Figure 3, the new VLP encoders, $E_{new}^I$ and $E_{new}^T$, and $\phi$ are trained through $\mathcal{L}_X$ in an end-to-end manner. Ultimately, $\bar{v}_{new}$ and $\bar{w}_{new}$ become cross-modal backward-compatible with existing old gallery embeddings, $v_{old}$ and $w_{old}$, as all embeddings are distributed in the compatible space through $\phi$.

**Efficient Training.** The efficiency of XBT can be attributed to two design choices that *avoid*: (1) any dependence on the old model when conducting XBT, and (2) updating the new VLP model off the shelf. We have already seen how $\phi$ can effectively achieve (1). (2) is simply facilitated by applying small-sized additional parameters, namely Soft prompt Lester et al. (2021) and LoRAHu et al. (2021).

To be more specific, we incorporate the concept of soft prompt tuning, as outlined in Lester et al. (2021); Jia et al. (2022). Soft prompting is applied to the vision encoder $E_{new}^I$ by adding trainable prompts as input, along with image patch tokens. We exclude soft prompt tuning on the text side, as

we use the text-only pretrained module $\phi$, which requires maintaining the original distribution of text embeddings. We adopt the LoRA strategy Hu et al. (2021) for the new VLP model's encoders $E_{new}^I$ and $E_{new}^T$ to avoid modifying original parameters. These two factors, in conjunction with $\phi$, offer an on-off solution: we can retrieve old samples using additional parameters, and we can easily revert to the original model by removing these parameters for new-to-new retrieval.

Further, regarding $\phi$ during XBT (Sec. 3.3), we only fine-tune the parameters of layer normalization. This is done with the aim of preserving the knowledge learned during the text-only pretraining stage (Sec. 3.2) and facilitating easy adaptation towards the image-text joint representation space.

In the end, our XBT framework offers a parameter-efficient solution that enables rapid convergence with fewer training iterations (a single epoch is sufficient); maintains the new model's power; and requires far fewer supervised training samples than the scale of VLP model pretraining (approximately 1% of the level required to train CLIP Radford et al. (2021) from scratch, which uses a 400M dataset, and 0.2% of the level required for LAION-2B based CLIP models Schuhmann et al. (2022), which utilize a 2B dataset). All these help to avoid any full-scale retraining of the new VLP models, paving the way for XBT to simply use new VLP models off-the-shelf as mentioned.

## 4 EXPERIMENTS

The evaluation of the proposed cross-modal backward compatibility includes image-text zero-shot retrieval. Pretrained VLP models are used as our baseline, with the goal to assess performance in a zero-shot environment by tuning a given pretrained new VLP model that is stronger than the old model. The results highlight the potential of XBT, showing improved performance when old model outputs are matched with the XBT-learned new model.

### 4.1 SETUP

**Model Training.** Our XBT process involves two distinct training stages: text-only pretraining (See Sec. 3.2) and image-text supervised training (See Sec. 3.3). For the text-only pretraining stage, we utilize the text samples from a subset of the 115M filtered web-collected image-text paired dataset from BLIP Li et al. (2022a), comprising around 67M available pairs (58.2% of the total). We construct a text corpus $D_T$ in Eqn. 3 using these text samples. For the subsequent image-text supervised training stage, we use a smaller subset of 4M image-text pairs from the same dataset to construct a supervised dataset $D$ in Eqn. 5. It is important to note that these subsets are significantly smaller than the dataset scale (400M, 2B or more) used to build CLIP models Radford et al. (2021); Schuhmann et al. (2022).

We utilize 8-A100-80GB GPUs to train and evaluate the models. For the text-only pretraining stage in Sec. 3.2, the batch size is set to 8,192 (1,024 batch per GPU) and during this stage, the entire set of trainable parameters of $\phi$ are trained while both VLP text encoders are fixed. Moving on to the image-text supervised training in Sec. 3.3, the batch size is reduced to 1,024 (128 batch per GPU), and in this stage, layer normalization is set to be the only trainable component for all VLP image and text encoders with $\phi$. Temperature hyper-parameters $\tau_{pre}$, $\tau_X$, and $\tau_N$ are fixed at 2.6592. We employ the AdamW optimizer Loshchilov & Hutter (2017) with a fixed learning rate of 1e-4 and a weight decay of 0.01. For soft prompts, we use 10 prompts and apply 100 times larger learning rate, 1e-2. The training iteration is determined by the dataset size, and the entire pipeline is trained for a *single epoch*. For image augmentation, we begin with a random resized crop, adjusting the image scale between 0.5 and 1.0. Additionally, we apply a random horizontal flip and make random adjustments to the image's contrast, brightness, and sharpness. To incorporate different perspectives and angles, we modify the image's translation and rotation. The pretrained weights provided by HuggingFace[1] Wolf et al. (2020) are applied to baseline VLP models. We employ as: `openai/clip-vit-base-patch32`, `openai/clip-vit-large-patch14`, `laion/CLIP-ViT-H-14-laion2B-s32B-b79K`.

**Model Evaluation.** We validate the effectiveness of XBT with a benchmark comprising three popular image-text paired datasets for cross-modal retrieval evaluation. The first is the *nocaps* dataset

---

[1]https://huggingface.co/models

Table 1: Cross-modal retrieval results on *nocaps*. Note that for comparison, the B32/L14 case should be compared with B32, and the L14/H14 case should be compared with L14, respectively.

| Old Model / New Model | Method | Text Query$(w, \bar{w})$/Image Gallery$(v)$ | | | | | Image Query$(v, \bar{v})$/Text Gallery$(w)$ | | | | |
|---|---|---|---|---|---|---|---|---|---|---|---|
| | | Case | R@1 | R@5 | R@10 | R@50 | Case | R@1 | R@5 | R@10 | R@50 |
| *Original* | | | | | | | | | | | |
| CLIP-ViT-B32 | - | $w/v$ | 45.14 | 74.98 | 84.51 | 96.00 | $v/w$ | 71.38 | 92.02 | 96.33 | 99.67 |
| CLIP-ViT-L14 | - | $w/v$ | 47.77 | 76.50 | 85.13 | 95.95 | $v/w$ | 73.29 | 93.24 | 97.40 | 99.78 |
| *Cross-modal Backward Compatible Training* | | | | | | | | | | | |
| CLIP-ViT-B32 / CLIP-ViT-L14 | *Full-tune* | $\bar{w}_{new}/v_{old}$ | 41.60 | 73.54 | 84.26 | 96.45 | $\bar{v}_{new}/w_{old}$ | 63.76 | 87.00 | 93.51 | 99.09 |
| | *LoRA-only* | | 43.40 | 74.66 | 84.94 | 96.73 | | 68.82 | 90.87 | 96.00 | 99.76 |
| | *Base* | | 43.48 | 74.87 | 85.04 | 96.74 | | 69.78 | 91.31 | 96.02 | 99.78 |
| | XBT | | **48.02** | **79.00** | **88.21** | **97.66** | | **75.02** | **93.27** | **97.31** | **99.91** |
| | *Full-tune* | $\bar{w}_{new}/\bar{v}_{new}$ | 48.92 | 79.31 | 88.26 | 97.64 | $\bar{v}_{new}/\bar{w}_{new}$ | 61.22 | 85.89 | 92.62 | 99.04 |
| | *LoRA-only* | | 55.44 | 83.82 | 91.24 | 98.34 | | 70.11 | 91.13 | 96.49 | 99.73 |
| | *Base* | | 56.18 | 84.31 | 91.44 | 98.37 | | 71.07 | 91.44 | 96.22 | 99.76 |
| | XBT | | **63.48** | **88.61** | **93.98** | **98.86** | | **77.22** | **94.56** | **97.91** | **99.89** |
| CLIP-ViT-L14 / CLIP-ViT-H14 | *Full-tune* | $\bar{w}_{new}/v_{old}$ | 45.18 | 77.82 | 85.34 | 97.65 | $\bar{v}_{new}/w_{old}$ | 66.38 | 88.55 | 93.59 | 98.90 |
| | *LoRA-only* | | 47.00 | 76.94 | 86.02 | 97.63 | | 72.04 | 92.42 | 96.58 | 99.67 |
| | *Base* | | 47.06 | 77.15 | 86.12 | 97.64 | | 73.00 | 92.86 | 96.60 | 99.69 |
| | XBT | | **51.14** | **80.82** | **88.93** | **98.72** | | **76.62** | **94.11** | **97.58** | **99.84** |
| | *Full-tune* | $\bar{w}_{new}/\bar{v}_{new}$ | 54.39 | 82.26 | 89.06 | 98.95 | $\bar{v}_{new}/\bar{w}_{new}$ | 65.57 | 87.52 | 93.51 | 99.13 |
| | *LoRA-only* | | 60.91 | 86.77 | 92.04 | 98.65 | | 74.46 | 92.73 | 97.38 | 99.82 |
| | *Base* | | 61.65 | 87.26 | 92.24 | 98.68 | | 75.42 | 93.04 | 97.11 | 99.85 |
| | XBT | | **66.47** | **90.21** | **94.92** | **99.05** | | **80.02** | **96.11** | **98.87** | **100.0** |

Agrawal et al. (2019), which offers a diverse distribution of object names and facilitates a more detailed analysis. We employ the validation split of this dataset, which consists of images each paired with 10 relevant textual captions, totaling 4,500 images and 45,000 captions.

To evaluate XBT at a larger scale, we also include the Flickr Huiskes & Lew (2008) and COCO Lin et al. (2014) datasets, which consist of images that are each paired with 5 relevant textual captions. For the Flickr dataset, we use the entire dataset, encompassing 31,783 images and 158,915 captions. For the COCO dataset, we use the validation split, which includes 35,136 images and 175,680 captions. We notate the two datasets as *Flickr-31K* and *COCO-35K* respectively for the remainder of the paper.

For retrieval, we utilize all the samples in each dataset for both query and gallery, following the cross-modal benchmark used in Radford et al. (2021); Li et al. (2022a). For evaluation purpose, we adopt recall scores at top K retrieval results (R@K, %) to estimate the cross-modal backward compatibility (See Eqn. 2).

**Implementation Details.** In our work, we primarily use the popular CLIP models Radford et al. (2021), based on a Transformer Vaswani et al. (2017) backbone, as our baseline VLP models. The models, listed in ascending order of scale and performance, are: CLIP-ViT-B32, CLIP-ViT-L14, and CLIP-ViT-H14. For simplicity, we will refer to them as B32, L14, and H14, respectively. Notably, while B32 and L14 are trained with the same dataset at 400M scale, H14 is trained with a larger dataset at 2B scale. We apply additional LoRA Hu et al. (2021) parameters for each new model encoder configured as follows: LoRA$_{\alpha}$ = 16, rank = 16, and dropout = 0.1. The proposed projection module $\phi$ consists of three Linear layers with layer normalization (LN) Ba et al. (2016) and the GELU non-linearity function Hendrycks & Gimpel (2016). Detailed architecture is as follows: $\texttt{Linear} - \texttt{LN} - \texttt{GELU} - \texttt{Linear} - \texttt{LN} - \texttt{GELU} - \texttt{Linear}$. Dropout is not applied as it was found to degrade performance empirically. The intermediate hidden dimension of the Linear layer is set to be four times the dimensionality of the output embedding.

## 4.2 MAIN RESULTS

**Baseline Comparisons.** To simplify notation, we use $w/v$ to denote the retrieval results obtained using $w$ as the query embeddings and $v$ as the gallery embeddings. We establish two protocols: assessing cross-modal backward compatibility with $new/old$ ($\bar{w}_{new}/v_{old}$, $\bar{v}_{new}/w_{old}$), and maintaining the new VLP's original performance with $new/new$ ($\bar{w}_{new}/\bar{v}_{new}$, $\bar{v}_{new}/\bar{w}_{new}$). All of the above are similarly applied to $v/w$.

Table 2: Cross-modal retrieval results on *Flickr-31K*.

| Old Model / New Model | Method | Text Query $(w, \bar{w})$/Image Gallery$(v)$ | | | | | Image Query $(v, \bar{v})$/Text Gallery$(w)$ | | | | |
|---|---|---|---|---|---|---|---|---|---|---|---|
| | | Case | R@1 | R@5 | R@10 | R@50 | Case | R@1 | R@5 | R@10 | R@50 |
| *Original* | | | | | | | | | | | |
| CLIP-ViT-B32 | - | $w/v$ | 21.54 | 41.24 | 50.72 | 72.69 | $v/w$ | 40.35 | 64.42 | 73.39 | 89.91 |
| *Cross-modal Backward Compatible Training* | | | | | | | | | | | |
| CLIP-ViT-B32 / CLIP-ViT-L14 | *Base* | $\bar{w}_{new}/v_{old}$ | 19.02 | 37.25 | 46.63 | 69.32 | $\bar{v}_{new}/w_{old}$ | 36.74 | 60.61 | 70.22 | 88.34 |
| | XBT | | **22.38** | **42.71** | **52.41** | **74.80** | | **42.47** | **66.04** | **74.87** | **90.79** |
| | *Base* | $\bar{w}_{new}/\bar{v}_{new}$ | 32.29 | 55.18 | 64.68 | 83.60 | $\bar{v}_{new}/\bar{w}_{new}$ | 36.10 | 59.80 | 69.95 | 89.01 |
| | XBT | | **39.59** | **62.89** | **71.75** | **88.13** | | **43.50** | **68.50** | **77.99** | **93.21** |
| CLIP-ViT-B32 / CLIP-ViT-H14 | *Base* | $\bar{w}_{new}/v_{old}$ | 20.27 | 39.17 | 48.59 | 70.67 | $\bar{v}_{new}/w_{old}$ | 35.53 | 59.44 | 69.06 | 87.67 |
| | XBT | | **23.70** | **44.47** | **54.19** | **76.03** | | **41.09** | **65.18** | **74.32** | **90.16** |
| | *Base* | $\bar{w}_{new}/\bar{v}_{new}$ | 32.94 | 55.94 | 65.26 | 83.77 | $\bar{v}_{new}/\bar{w}_{new}$ | 34.81 | 59.55 | 69.96 | 88.81 |
| | XBT | | **40.42** | **63.90** | **72.74** | **88.47** | | **46.35** | **71.18** | **80.01** | **94.08** |

To the best of our knowledge, since this paper presents the first method that aims to solve the XBT problem, there are no direct previous works to compare to. Nevertheless, we compare our XBT against three baselines: *Full-tune*, *LoRA-only*, and *Base*, which we elaborate on below.

*Naïve solution - Direct backward contrastive learning:* The pretrained $\phi$ in Eqn. 3, which maps the new embeddings to the old via text based contrastive learning, constitutes a major contribution of XBT. We therefore would like to compare with a solution that does not utilize such a pretrained $\phi$. A straightforward solution for this is to fine-tune the new VLP encoders ($E_{new}^I$ and $E_{new}^T$) to be compatible with the old ones by minimizing the following loss:

$$\mathcal{L}_{Direct} = \mathbb{E}_{x,t \sim D}[\mathcal{L}_c(\bar{v}_{new}, w_{old}; \tau_N) \\ + \mathcal{L}_c(\bar{w}_{new}, v_{old}; \tau_N)] \tag{6}$$

where $\mathcal{L}_c(\cdot, \cdot)$ is the same contrastive loss used in Eqn. 5, and $\tau_N$ is a temperature. Here, we apply a randomly initialized $\phi$ with the same configuration as XBT for projection, as per Eqn. 4, to generate cross-modal backward compatible embeddings, $\bar{w}$ and $\bar{v}$.

*Different training options:* Based on Eqn. 6, we consider multiple training setups: (1) full fine-tuning of all trainable components (*Full-tune*), (2) tuning LoRA parameters-only (*LoRA-only*), and (3) starting from *LoRA-only*, adding an extra learnable prompt (*Base*). Models produced by *Base* would therefore be akin to XBT without the pretrained $\phi$. For these, we train the $E_{new}^I$, $E_{new}^T$ and $\phi$ using $\mathcal{L}_{Direct}$ with 4M image-text pairs. Note that, XBT only trains layer-normalization layers including pretrained $\phi$, however, *Full-tune*, *LoRA-only* and *Base* setups are training entire parameters of $\phi$ since it is not trained before. The results of these baselines are presented in Table 1, tested with *nocaps* dataset. The performance of the original VLP models is also reported for a clear comparison. We highlight our XBT method.

In the context of cross-modal backward compatibility, where recall scores of $\bar{w}_{new}/v_{old}$ and $\bar{v}_{new}/w_{old}$ should be higher than those of $w/v$ and $v/w$ of old model, respectively, *Full-tune* significantly underperforms compared to the old VLP's. This indicates that full fine-tuning is not an appropriate solution. While both *LoRA-only* and *Base* improve performance over *Full-tune*, they still fall short of the old VLP's performance and fail to meet the criterion. In contrast, XBT significantly outperforms these baselines and even surpasses the old VLP's performance, satisfying the cross-modal backward compatibility for all recall metrics (see Eqn. 2).

In terms of maintaining new VLP's performance, where recall scores of $\bar{w}_{new}/v_{new}$ and $\bar{v}_{new}/w_{new}$ could be similar to those of $w/v$ and $v/w$ of new model, respectively, all baselines show decent performance in $\bar{w}_{new}/v_{new}$ case but only XBT achieves better performance than new VLP model in $\bar{v}_{new}/w_{new}$ case. These results support that XBT not only maintains the performance of the new VLP model but also enhances it in certain cases. The superior performance of XBT in both cases underscores its promise as a robust solution for cross-modal retrieval tasks.

**Large Scale Retrieval.** In this experiment, we assess the performance of cross-modal backward compatibility in more practical use cases. We use larger datasets, namely *Flickr-31K* and *COCO-35K*, as shown in Tables 2 and 3. These datasets provide more than six times the number of samples

Table 3: Cross-modal retrieval results on *COCO-35K*.

| Old Model / New Model | Method | Text Query$(w,\bar{w})$/Image Gallery$(v)$ | | | | | Image Query$(v,\bar{v})$/Text Gallery$(w)$ | | | | |
|---|---|---|---|---|---|---|---|---|---|---|---|
| | | Case | R@1 | R@5 | R@10 | R@50 | Case | R@1 | R@5 | R@10 | R@50 |
| *Original* | | | | | | | | | | | |
| CLIP-ViT-B32 | - | $w/v$ | 14.44 | 30.20 | 38.96 | 62.30 | $v/w$ | 28.62 | 50.17 | 59.67 | 80.22 |
| *Cross-modal Backward Compatible Training* | | | | | | | | | | | |
| CLIP-ViT-B32 / CLIP-ViT-L14 | *Base* | $\bar{w}_{new}/v_{old}$ | 12.97 | 28.14 | 36.48 | 59.78 | $\bar{v}_{new}/w_{old}$ | 26.32 | 46.91 | 56.54 | 77.86 |
| | XBT | | **15.55** | **32.27** | **41.34** | **65.11** | | **30.73** | **52.58** | **62.30** | **81.83** |
| | *Base* | $\bar{w}_{new}/\bar{v}_{new}$ | 22.16 | 41.65 | 50.98 | 73.06 | $\bar{v}_{new}/\bar{w}_{new}$ | 27.26 | 48.16 | 57.80 | 79.57 |
| | XBT | | **27.61** | **48.69** | **58.08** | **78.83** | | **32.92** | **55.37** | **64.89** | **84.67** |
| CLIP-ViT-B32 / CLIP-ViT-H14 | *Base* | $\bar{w}_{new}/v_{old}$ | 13.88 | 29.55 | 38.23 | 61.57 | $\bar{v}_{new}/w_{old}$ | 26.19 | 46.75 | 56.30 | 77.96 |
| | XBT | | **15.82** | **32.79** | **41.84** | **65.45** | | **30.33** | **51.28** | **59.97** | **81.66** |
| | *Base* | $\bar{w}_{new}/\bar{v}_{new}$ | 23.20 | 43.22 | 52.70 | 74.68 | $\bar{v}_{new}/\bar{w}_{new}$ | 27.31 | 48.56 | 58.60 | 80.09 |
| | XBT | | **28.14** | **49.48** | **58.88** | **79.32** | | **34.54** | **57.29** | **66.79** | **85.92** |

Table 4: Ablation study tested on *nocaps*, with B32 as *old* and L14 as *new* model.

| Method | $\bar{w}$(text)/$v$(image) | | $\bar{v}$(image)/$w$(text) | |
|---|---|---|---|---|
| | R@1 | R@10 | R@1 | R@10 |
| (a) Baseline | 48.02 | 79.00 | 75.02 | 93.27 |
| (b) w.o noise | 47.24 | 78.05 | 73.31 | 92.49 |
| (c) $0.25 \times D_T$ | 46.89 | 77.91 | 73.20 | 92.18 |
| (d) $0.5 \times D_T$ | 47.68 | 78.00 | 73.89 | 92.00 |
| (e) $2 \times D$ | 47.63 | 78.69 | 74.98 | 93.98 |
| (f) $0.5 \times D$ | 46.80 | 77.89 | 71.69 | 92.00 |
| (g) $D$=CC3M | 46.44 | 77.76 | 71.33 | 91.84 |
| (h) Image-only | 45.32 | 76.92 | 74.02 | 92.35 |

Table 5: Cross-modal retrieval results on *nocaps*, using BLIP-Base and BLIP-Large.

| Method | Text Query/Image Gallery | | | Image Query/Text Gallery | | |
|---|---|---|---|---|---|---|
| | Case | R@10 | R@50 | Case | R@10 | R@50 |
| *Original* | | | | | | |
| - | $w/v$ | 69.56 | 91.86 | $v/w$ | 85.44 | 97.89 |
| *Cross-modal Backward Compatible Training* | | | | | | |
| *Base* | $\bar{w}_{new}/v_{old}$ | 67.61 | 91.56 | $\bar{v}_{new}/w_{old}$ | 81.42 | 97.38 |
| XBT | | **69.90** | **92.52** | | **85.83** | **98.04** |
| *Base* | $\bar{w}_{new}/\bar{v}_{new}$ | 69.02 | 90.57 | $\bar{v}_{new}/\bar{w}_{new}$ | 79.29 | 95.98 |
| XBT | | **73.17** | **93.27** | | **89.16** | **98.88** |

typically used in image-text retrieval literature Wan et al. (2014); Chen et al. (2023), which make the retrieval process more challenging. We use the second-best performing *Base* method from Table 1 as a basis for comparison. To validate the applicability of XBT across various VLP models, we utilize three CLIP baselines: B32, L14, and H14. Additionally, we explore different combinations, such as B32/L14 and B32/H14, to further analyze the effectiveness of XBT, beyond the comparisons made in Table 1 (B32/L14 and L14/H32).

When comparing B32/L14 with B32/H14, it is observed that every retrieval cases of improved models (L14 and H14) successfully maintain cross-modal backward compatibility, regardless of their scale. Especially, retrieval performance of B32/H14 is better than that of B32/L14, which demonstrates XBT's ability to preserve and align original model's power with old gallery.

Upon examining the results, we consistently observe that XBT outperforms the *Base* model across all instances and both retrieval scenarios, often by a significant margin. Furthermore, XBT maintains the performance of new VLPs better than *Base*, achieving even larger margins for $\bar{w}_{new}/\bar{v}_{new}$ and $\bar{v}_{new}/w_{old}$. In summary, XBT consistently demonstrates robust and superior retrieval performance, even in large-scale scenarios.

## 4.3 FURTHER ANALYSIS

**Ablation Study.** To validate XBT, we conduct further analysis as shown in Table 4. By comparing (a) and (b), we observe that introducing noise during $\phi$ training aids in generalization. The comparison between (c) and (d) demonstrates that the scale of $D_T$ is important, supporting our assumption that a sufficient number of text samples can help build a robust $\phi$. (e) outperforms (a), confirming that utilizing more image-text pairs can enhance XBT. The lower performance of (f) also aligns with the notion that the number of image-text pairs is crucial. In (g), when we replace $D$ with CC3M Sharma et al. (2018), from which we can obtain around 2.4M, the performance is similar to (f), suggesting that XBT can be effectively applied with other datasets. In (h), we replace text-only pretraining with image-only pretraining of the same scale and observe that text-only pretraining performs better in both retrieval scenarios, demonstrating its efficiency and superiority.

Table 6: Continual learning scenario experimental results with XBT on *nocaps*. Note that for H14 training, we use L14, which has been previously adapted with B32, and H14 never encounters B14.

| Old Model / New Model | Text Query$(w, \bar{w})$/Image Gallery$(v)$ | | | | | Image Query$(v, \bar{v})$/Text Gallery$(w)$ | | | | |
|---|---|---|---|---|---|---|---|---|---|---|
| | Case | R@1 | R@5 | R@10 | R@50 | Case | R@1 | R@5 | R@10 | R@50 |
| CLIP-ViT-B32 / | $\bar{w}_{new}/v_{old}$ | 47.68 | 78.00 | 88.00 | 97.50 | $\bar{v}_{new}/w_{old}$ | 73.89 | 92.00 | 97.22 | 99.86 |
| CLIP-ViT-L14 | $\bar{w}_{new}/\bar{v}_{new}$ | 62.22 | 86.18 | 93.00 | 98.12 | $\bar{v}_{new}/\bar{w}_{new}$ | 76.27 | 92.90 | 96.76 | 99.11 |
| CLIP-ViT-L14 / | $\bar{w}_{new}/v_{old}$ | 50.97 | 80.68 | 88.98 | 97.72 | $\bar{v}_{new}/w_{old}$ | 77.38 | 94.82 | 97.80 | 99.84 |
| CLIP-ViT-H14 | $\bar{w}_{new}/\bar{v}_{new}$ | 66.09 | 89.78 | 94.77 | 99.01 | $\bar{v}_{new}/\bar{w}_{new}$ | 80.60 | 96.36 | 98.71 | 99.96 |
| CLIP-ViT-B32 / | $\bar{w}_{new}/v_{old}$ | 49.14 | 79.88 | 88.59 | 97.67 | $\bar{v}_{new}/w_{old}$ | 74.84 | 93.73 | 97.22 | 99.80 |
| CLIP-ViT-H14 | $\bar{w}_{new}/\bar{v}_{new}$ | 65.95 | 89.68 | 94.68 | 98.99 | $\bar{v}_{new}/\bar{w}_{new}$ | 80.96 | 96.00 | 98.71 | 99.96 |

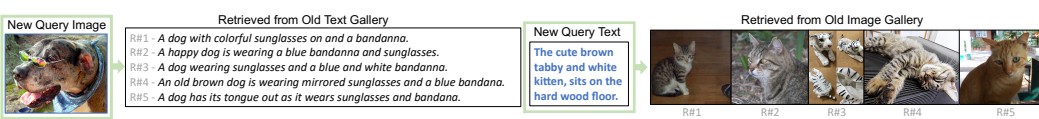

Figure 4: *New* query vs. *Old* gallery retrieval results on *nocaps*. B32 as old, and L14 as new model.

**Different VLP models.** To further explore the capacity of the VLP model architecture's generalization of XBT, we evaluate it using BLIP Li et al. (2022a) Base and Large models. We employ checkpoints of `Salesforce/blip-itm-base-coco` and `Salesforce/blip-itm-large-coco` from the Huggingface library. We apply XBT on old and new BLIP models in the same fashion with our CLIP applications, and the results appear in Table 5. From the results, we confirm that XBT provides cross-modal backward-compatibility to the BLIP models too.

**Continual learning.** In line with the literature on BT works Shen et al. (2020); Ramanujan et al. (2022), we set up a continual learning scenario for a sequence of model updates (old model B32, new model L14, and better new model H14) and present the retrieval results in Table 6. Initially, we apply XBT to L14 to ensure compatibility with B32. Subsequently, we apply XBT to H14 to ensure compatibility with the previously learned L32, which is already compatible with B32. For this process, we divide both the image-text pairs and the text-only pretraining train sets into two halves, using each separately for each case. A comparison with the original results reported in Table 6 confirms that our XBT performs well in the continual learning scenario either. It achieves cross-modal backward compatibility while leveraging the power of the improved new model.

**Qualitative Results.** We facilitate retrieval by utilizing new query embeddings and old gallery embeddings. The results in Figure 4 demonstrate accurate cross-modal backward-compatible retrieval.

## 5 DISCUSSION & CONCLUSION

**Potential Broader Impact and Limitation.** This paper highlights our efforts to enhance the field of Machine Learning, particularly in the area of multi-modal embedding-based representation learning. Although our work may have societal implications, we do not believe there are any that require specific emphasis in this context. A potential limitation of the XBT system is that, despite the efficient learning approach reducing the need for image-caption pairs, its performance may still be limited by the quality, diversity, and representational richness of the data during training.

**Conclusion.** In this paper, we introduced Cross-modal Backward-compatible Training (XBT), a novel task for cross-modal retrieval that focuses on the compatibility between image and text embeddings of different Vision-Language Pretraining (VLP) models. We proposed an efficient solution using a text-only pretrained projection module, $\phi$, to align the new model's embeddings with those of the old model, thereby enhancing training efficiency. By integrating parameter-efficient training schemes into the XBT framework, we were able to accelerate the model's training while preserving the original VLP's zero-shot capabilities. Our approach, demonstrated on various cross-modal benchmarks, effectively builds cross-modal retrieval systems without backfilling, offering an efficient and environmentally friendly solution in response to the VLP improvements.

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

# A  APPENDIX

## A.1  DATASET EXAMPLES, MORE VISUALIZATION

In Figure 5, we use a t-SNE map to examine the actual distribution of embeddings in the VLP space. It's evident that the image and text embeddings are distinct. Furthermore, the intra-distribution within both image and text embeddings is similar, suggesting that they are supposed to *mirror* each other.

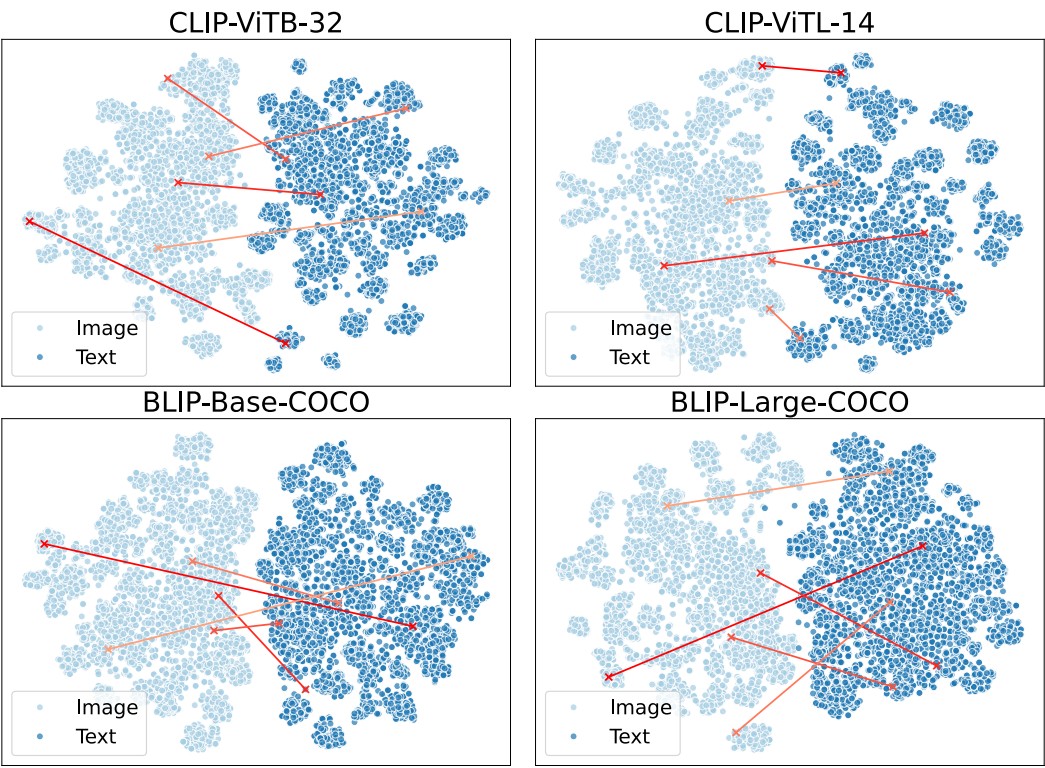

Figure 5: A tSNE visualization of 5,000 paired image-text embeddings from COCO Lin et al. (2014) dataset, using two different CLIP models Radford et al. (2021), and two different BLIP models Li et al. (2022a). Five pairs are marked as examples. The distinct distributions of image and text samples in each VLP space are observed.

## A.2  FURTHER ANALYSIS & DISCUSSION

Table 7: Computational analysis on baselines. We evaluate with B32 as *old* and L14 as *new* model.

| Method | Training Time (h) | Trainable Parameters (M) | Memory Load (GB) | Number of Samples (M) |
|---|---|---|---|---|
| Text-only Pretraining | 1.55 | 6.82 | 0.61 | 67 |
| *Full-tune* | 5.71 | 434.45 | 1.13 | 4 |
| *LoRA-only* | 5.42 | 8.34 | 1.13 | 4 |
| *Base* | 5.66 | 8.35 | 1.13 | 4 |
| XBT | 2.84 | 8.36 | 0.84 | 4 |

**Computational Analysis.**  In Table 7, we calculate the required training cost for each baseline. Despite XBT handling a larger number of training samples, the total training time (Text-only pretraining + XBT) is less than that of the other methods. Furthermore, since XBT does not utilize the old VLP model during training, it significantly reduces the memory load.

Table 8: Zero-shot classification results on ImageNet Russakovsky et al. (2015), ImageNet-R Hendrycks et al. (2021), and ImageNet-Sketch Wang et al. (2019). $\bar{w}$ and $\bar{v}$ are used to compute scores, and accuracy (%) is metric.

| Method | ImageNet | ImageNet-R | ImageNet-Sketch |
|---|---|---|---|
| CLIP-ViTB-32 | 55.23 | 40.66 | 35.53 |
| CLIP-ViTL-14 | 66.63 | 62.30 | 52.52 |
| XBT trained by 4M | 55.44 | 59.21 | 45.67 |
| XBT trained by 8M | 55.91 | 61.27 | 47.02 |
| XBT trained by 16M | 57.99 | 63.53 | 48.69 |

**Research question: Zero-shot Classification.** As we incorporate VLP models, an intriguing research question emerges: How do VLP models, fine-tuned with XBT, perform as zero-shot classifiers? To investigate this, we conduct a zero-shot classification using the text prompt *'a photo of class name'*. As demonstrated in Table 8, XBT outperforms the old VLP in classification performance, though it falls short of the new VLP. Interestingly, we observe that as the number of supervised training samples increases, so does the classification performance. This suggests the potential for XBT-tuned models to function as zero-shot classifiers given sufficient training samples. This opens up a new research direction towards not only achieving backward compatibility, but also comparable performance to zero-shot classifiers.

