# OpenReview forum: "Towards Cross-modal Backward-compatible Representation Learning for Vision-Language Models"
_ICLR.cc/2025/Conference — Submitted to ICLR 2025_

### Official Review · Reviewer_jh9S · 2024-10-30

**Soundness:** 3
**Presentation:** 3
**Contribution:** 3
**Rating:** 6
**Confidence:** 4

**Summary:**

The authors propose a novel approach called Cross-modal Backward-compatible Training (XBT), which extends the concept of Backward-compatible Training (BT) from vision-only to cross-modal retrieval, specifically for aligning Vision-Language Pretraining (VLP) models like CLIP.   A text-only pre-trained projection module, ϕ, is proposed to map new model embeddings to old model embeddings. This module is trained solely with text data, reducing the need for image-text pairs. The paper utilizes parameter-efficient training strategies that improve efficiency and preserve the new model's knowledge by avoiding modifications to it. Experiments on cross-modal retrieval datasets show XBT's effectiveness and its potential to enable backfill-free upgrades when new VLP models emerge.

**Strengths:**

1. The targeting problem is interesting. And it is might be useful for the real applications.

2. The proposed method achieved expecting performance.

**Weaknesses:**

1. It seems that the motivation of the text-only pre-training stage is not very clear.

**Questions:**

1. Why is the text-only pre-training necessary?

2. Why preventing overfitting in the text-only pre-training stage is needed ? Are there any explanations for that?

---

> ### Author Response · Authors · 2024-11-21
>
> We appreciate you for recognizing our novel XBT framework, which extends Backward-compatible Training (BT) to cross-modal retrieval by aligning Vision-Language Pretraining models like CLIP using a text-only pre-trained projection module, leveraging parameter-efficient training to reduce the reliance on image-text pairs and preserve the new model's knowledge. We address your comments in detail below.
>
> **w1:** The motivation for the text-only pre-training stage lies in addressing the impracticality of requiring large-scale supervised image-text pairs for cross-modal backward-compatible training. Instead, we propose using text-only data to estimate the embedding distribution within the Vision-Language Pretraining space. This approach assumes that the intra-modal distribution of text embeddings mirrors that of images, allowing us to align new and old embeddings effectively through a pretrained projection module, φ. This solution significantly reduces the reliance on image-text pairs, making the training process more efficient and feasible for large-scale systems. We will clarify this motivation more explicitly in the final version of the paper.
>
> **q1:** The text-only pre-training stage is necessary to address the scalability and feasibility challenges of cross-modal backward-compatible training. Obtaining large-scale supervised image-text pairs similar to those used in the original VLP model training is often impractical due to resource constraints and data unavailability. By relying solely on text data, we estimate the embedding distribution within the VLP space and align new and old embeddings using the pretrained projection module, φ. This approach significantly reduces the reliance on image-text pairs, making the training process both efficient and scalable.
>
> **q2:** Preventing overfitting in the text-only pre-training stage is essential to ensure that the pretrained projection module, φ, generalizes well to unseen data and maintains its ability to align embeddings across modalities. Overfitting could lead to φ memorizing specific text embeddings rather than learning a robust mapping between the distributions of new and old embeddings. This would undermine the assumption that text embeddings mirror image embeddings, reducing the effectiveness of the alignment process. To address this, we use robust training techniques, such as contrastive learning and moderate noise injection, to ensure φ captures the broader distribution patterns rather than overfitting to the training data.

---

> ### Author Response · Authors · 2024-11-25
>
> Dear reviewer jh9S,
>
> We appreciate your valuable comments on our paper. We have prepared a rebuttal with an updated manuscript and tried our best to address your concerns. We notice that the author-reviewer discussion period is coming to an end, and we are willing to answer any unresolved or further questions that you may have regarding our rebuttal if time is allowed.
>
> If our rebuttal has addressed your concerns, we would appreciate it if you would let us know your final thoughts. Additionally, we will be happy to answer any further questions regarding the paper. Thank you for your time and consideration.

---

### Official Review · Reviewer_vB1V · 2024-10-31

**Soundness:** 2
**Presentation:** 2
**Contribution:** 3
**Rating:** 5
**Confidence:** 4

**Summary:**

This paper presents a cross-modal backward-compatible training (XBT) method that enables the deployment of new multimodal encoders while maintaining compatibility with existing retrieval systems. The key innovation lies in its approach to align a single modality to potentially achieve cross-modal alignment, offering a practical solution for real-world system updates. The work addresses an important practical challenge in maintaining backward compatibility while upgrading multimodal systems.

**Strengths:**

1.	Comprehensive Experimentation: The paper demonstrates strong empirical validation through extensive experiments across multiple models and datasets. This thorough evaluation effectively showcases the method's generalization capability and robustness across different scenarios.
2.	Novel Perspective: The authors present an innovative approach by proposing that aligning a single modality can potentially lead to cross-modal alignment. This insight offers a fresh perspective on handling multimodal compatibility issues and could inspire new research directions in the field.
3.	Practical Relevance: The work addresses a real-world challenge in deploying updated multimodal systems while maintaining compatibility with existing infrastructure, making it valuable for practical applications.

**Weaknesses:**

1.	Some citation formats are incorrect.
2.	Most performance tables are confusing, making it difficult to clearly understand the performance of both old and new retrieval systems.
3.	Lack of several analytical experiments, which are detailed in questions.

**Questions:**

1.	Can you show the similarity between new and old representations after φ's transformation for both image and text? Is image similarity lower than text similarity? What is the relationship between similarity and performance? How does noise magnitude in φ affect similarity?
2.	In Table 1, B32 is used as an old encoder model in the original column. Why does B32 show better performance after XBT training compared to the original B32? Can XBT improve the performance of old retrieval systems?
3.	Could the tables be reorganized into three parts to show the retrieval performance of:
-	Old systems
-	New systems
-	Proposed backward-compatible new systems

---

> ### Author Response · Authors · 2024-11-21
>
> We thank you for recognizing the comprehensive experimentation validating our method's generalization and robustness, the novel perspective of leveraging single-modality alignment for cross-modal compatibility, and the practical relevance of addressing real-world challenges in deploying updated multimodal systems. We respond to your comments in detail below.
>
> **w1:** We thank you for pointing this out. We will carefully review and correct the citation formats in the final version of the paper.
>
> **w2:** In backward-compatible representation learning for retrieval tasks, it is essential to compare the performance of new-to-new and new-to-old retrieval with old-to-old retrieval, as outlined in the constraints of Equation (2). By comparing these results in the tables, it can be observed that our XBT model successfully achieves backward compatibility in cross-modal retrieval scenarios. While we have made efforts to clearly present the performance of our model in the tables, we acknowledge the confusion and will provide further clarification and improve the presentation in the final version of the paper.
>
> **q1:** Following your suggestion, we measured the cosine similarity between the new model embeddings (CLIP-ViT-L14 with φ) and the old model embeddings (CLIP-ViT-B32) using 5,000 images and texts from the nocaps dataset. The average similarity is 0.867 for images and 0.889 for texts (with 1.0 being the maximum similarity). This indicates that, despite relying on text-only pretraining, the new embeddings for images are well-aligned with the old image embeddings. Higher similarity reflects that the new embeddings are effectively distributed close to the old embeddings, but if the similarity becomes too high, performance can degrade and approach the lower bound. Regarding noise magnitude in φ, it is only utilized during robust training and not during inference. If the noise magnitude is too high, it increases the distance between embeddings, and if too low, it results in overly close embeddings. We found that a standard deviation of 0.05 strikes an effective balance.
>
> **q2:** To clarify, in the retrieval scenarios of XBT, the query embeddings are always produced by the new model, while the gallery embeddings are precomputed using the old model (B32). The goal of XBT training is to make the query embeddings from the new model compatible with the old gallery embeddings. Thus, the old model itself (B32) remains unchanged, and only the new model is updated with LoRA parameters and an additional projection module (φ) to align with the precomputed old embeddings. XBT does not directly improve the old model but enhances compatibility between the new and old systems.
>
> **q3:** Following your suggestion, we will revise the tables to clearly separate and present the performance of old-to-old, new-to-new, new (proposed)-to-old, and new (proposed)-to-new (proposed) retrieval systems for better readability and understanding.

---

> ### Author Response · Authors · 2024-11-25
>
> Dear reviewer vB1V,
>
> We appreciate your valuable comments on our paper. We have prepared a rebuttal with an updated manuscript and tried our best to address your concerns. We notice that the author-reviewer discussion period is coming to an end, and we are willing to answer any unresolved or further questions that you may have regarding our rebuttal if time is allowed.
>
> If our rebuttal has addressed your concerns, we would appreciate it if you would let us know your final thoughts. Additionally, we will be happy to answer any further questions regarding the paper. Thank you for your time and consideration.

---

### Official Review · Reviewer_2yQn · 2024-11-04

**Soundness:** 3
**Presentation:** 3
**Contribution:** 2
**Rating:** 6
**Confidence:** 4

**Summary:**

The paper presents an approach to address the challenge of model upgrades in cross-modal retrieval systems by introducing Cross-modal Backward-compatible Training (XBT). This approach aims to align the embeddings of new Vision-Language Pretraining (VLP) models with those of the old models, thereby reducing the need for costly re-embedding of data, known as backfilling.

**Strengths:**

1. The concept of XBT is innovative and addresses a significant practical issue in the deployment of new models in real-world applications. The idea of using a text-only pretrained projection module to align embeddings is interesting and shows potential for efficiency gains.
2. The methodology is well-explained and technically sound. The authors have provided a clear outline of the training process, including the text-only pretraining and the cross-modal backward-compatible training stages.
3. The paper is well-supported by empirical evidence, with experiments on multiple benchmarks demonstrating the effectiveness of XBT in cross-modal retrieval tasks.

**Weaknesses:**

1. While the paper claims that XBT reduces the need for image-text pairs, the scalability of the approach in very large-scale systems with real-world data distributions is not fully explored. The author only used a source dataset from BLIP (which is a sythetic dataset), a subset from LAION400M will be benificial for this work.
2. The paper briefly mentions the impact of data quality on XBT performance. A more detailed analysis on how noisy or biased training data might affect the embedding alignment could provide deeper insights.
3. The computational analysis is limited to a comparison of training times and memory loads. A more comprehensive analysis, including the trade-offs between accuracy and computational resources, would be valuable.

**Questions:**

Please see weaknesses.

---

> ### Author Response · Authors · 2024-11-21
>
> We appreciate you for recognizing the innovation and practical significance of our XBT framework, the potential of our text-only pretrained projection module for efficiency gains, the clarity and technical soundness of our methodology, and the strong empirical evidence demonstrating its effectiveness across multiple benchmarks. We address your comments in detail below.
>
> **w1:** As shown in Table 4 (experiments c and d), we demonstrate the scalability of our approach with respect to text-only datasets and image-text pairs in (e and f). The results show that 67M text samples and 4M image-text pairs are sufficient to achieve the best performance, with no significant performance gains observed when increasing the number of samples in (d) and (f). This aligns with our goal of creating an "efficient" cross-modal backward-compatible system designed to train only LoRA parameters and a small projection module. While utilizing a dataset at the scale of LAION400M for training could potentially allow full training from scratch to build a new pretrained model, it would be highly inefficient and likely yield only minimal performance improvements. Instead, our approach prioritizes efficient text-only pretraining and LoRA tuning with significantly fewer image-text pairs, making it more practical and resource-efficient.
>
>
> **w2**: In Table 4 (experiment g), we evaluate our model using image-caption pairs from the CC3M dataset, which contains noisy data, including low-quality images and short captions that are sometimes weakly related to the images. As expected, the performance is degraded compared to the baseline dataset we used. To provide deeper insights, we will include detailed examples of the noisy data and reference related studies, such as [1], for further clarification in the final version.
>
> [1] Blip: Bootstrapping language-image pre-training for unified vision-language understanding and generation, Li et al, ICML2022
>
>
> **w3:** To address the trade-offs between accuracy and computational resources, we provide experimental results on the impact of batch size during training. Using the nocaps dataset for Text Query / Image Gallery retrieval, with CLIP-ViT-B32 as the old model and CLIP-ViT-L14 as the new model, we observe the following performance (R@1|R@5|R@10|R@50, respectively):
>
> Baseline (batch size 8,192): 48.02 | 79.00 | 88.21 | 97.66
> Batch size 1,024: 46.92 | 78.08 | 87.48 | 96.98
> Batch size 4,096: 47.86 | 78.34 | 88.06 | 97.34
> Batch size 16,384: 48.08 | 78.84 | 88.24 | 97.68
>
> The results indicate that the impact of batch size on XBT's performance is minimal when it exceeds a certain threshold (approximately 4,096, or 512 per GPU). We will include additional results and a more detailed analysis in the final version of the paper.

---

> ### Author Response · Authors · 2024-11-25
>
> Dear reviewer 2yQn,
>
> We appreciate your valuable comments on our paper. We have prepared a rebuttal with an updated manuscript and tried our best to address your concerns. We notice that the author-reviewer discussion period is coming to an end, and we are willing to answer any unresolved or further questions that you may have regarding our rebuttal if time is allowed.
>
> If our rebuttal has addressed your concerns, we would appreciate it if you would let us know your final thoughts. Additionally, we will be happy to answer any further questions regarding the paper. Thank you for your time and consideration.

---

### Official Review · Reviewer_nZUG · 2024-11-04

**Soundness:** 2
**Presentation:** 2
**Contribution:** 2
**Rating:** 5
**Confidence:** 4

**Summary:**

This paper introduces Cross-modal Backward-compatible Training (XBT), designed to address the challenges of upgrading vision-language models in cross-modal retrieval systems. Traditional methods require costly backfilling, where old embeddings must be re-computed with new models, but XBT aims to eliminate this necessity. The authors propose a text-only pretrained projection module that aligns new model embeddings with those of the old model, significantly reducing the need for extensive image-text pairs. This method enhances training efficiency and preserves the knowledge of the new model while allowing for effective fine-tuning. Experimental results demonstrate the effectiveness of XBT in various cross-modal benchmarks, offering a promising solution for backfill-free upgrades in retrieval systems.

**Strengths:**

- The motivation is meaningful. The paper seeks to avoid the time-consuming updating calculations on the embeddings of a cross-modal retrieval system when a new, improved model is introduced.

- The paper aims to develop a backfill-free cross-modal retrieval system and extends Backward-compatible Training from the vision domain to the cross-modal domain, potentially opening a new research direction.

- The effectiveness of the proposed method is supported by the experimental results.

**Weaknesses:**

- Poor formulation of the paper: There are incorrect citations in the paper. For example, in Lines 116 and 117, “(BT) was first introduced in the study Shen et al. (2020)” should be “(BT) was first introduced in the study (Shen et al. 2020).” Additionally, in Fig. 2, the captions refer to the wrong figures: "text-only pretraining" is on the left side, while XBT is on the right side, but the paper states "above" and "below."
- The backward-compatible representation seems counterintuitive. If the old model creates a flawed representational space, forcibly mapping the new, more powerful model to the old model's representation space could compromise the new model's representation. Could the authors discuss this issue?
- The baseline only considers the old model. As I understand it, the upper bound of a retrieval system when introducing a new, improved model should be based on the new model, while the old model serves as the lower bound.
- The retrieval performances of CLIP-ViT-B32 on Flickr and COCO are extremely low. According to the original CLIP paper, CLIP-ViT-L14 can achieve 88% R@1 on Flickr's text retrieval, while the performance of CLIP-ViT-B32 reported in this paper is 40.35%, which is over 40% lower than CLIP-ViT-L14. This lead to the experimental results unconvincing.

**Questions:**

Please refer to the weaknesses to check out the questions.

---

> ### Author Response · Authors · 2024-11-21
>
> We thank you for recognizing our meaningful motivation, the novel extension of Backward-compatible Training to cross-modal retrieval, and the strong experimental validation of our method. We address your thoughts point by point below.
>
> **w1:** We appreciate the reviewer pointing out these inaccuracies. We will carefully correct the citation formatting and revise the captions in Figure 2 to ensure clarity and consistency in the final version of the paper.
>
> **w2:**  Backward-compatible embeddings are essential for addressing the "backfilling" problem in scenarios where extremely large-scale old gallery embeddings already exist, as rebuilding the gallery with a new model can take several months. We have discussed this issue in detail in Lines 42–53 and refer the reviewer to other Backward-compatible Training (BT) works [1–3] for additional context. While BT may slightly compromise the new model's original performance, it enables retrieval with significantly better accuracy than the old model while minimizing performance degradation. Furthermore, we will supplement the paper by explicitly defining the cross-modal backward-compatible problem in the context of the constraints outlined in Equation (2) and demonstrating how our XBT model fulfills these constraints through experiments. Our results show that the XBT model achieves superior performance when using a new query with the new XBT model for retrieval, even with the old extensive gallery, compared to using the old model. This successfully addresses the challenge.
>
> [1] Towards backward-compatible representation learning, Shen et al, CVPR2020
> [2] Learning compatible embeddings, Zhang et al, ICCV2021
> [3] Forward compatible training for large-scale embedding retrieval systems, Ramanujan et al, CVPR2022
>
> **w3:** The primary reason we compare with the old model is that backward-compatible representation learning specifically aims to ensure compatibility between the new model's query embeddings and the old gallery embeddings, addressing the practical challenge of avoiding backfilling. The success of this compatibility is clearly demonstrated in Tables 1–5, where our XBT model significantly outperforms the old model in these scenarios. Furthermore, as shown in Table 1, the performance of XBT with new-to-new model retrieval not only surpasses the old model but also exceeds the expected upper bound, highlighting the robustness and advantages of our approach.
>
> **w4:** To clarify, our retrieval experiments on Flickr and COCO differ from those in the original CLIP paper. Specifically, we use a much more challenging setup with larger gallery sizes—31K and 35K images, respectively—compared to the 5,000 images used in CLIP's experiments, as detailed in Lines 349–353. This is why we explicitly named our setups Flickr-31K and COCO-35K to distinguish them from CLIP's original experiments. These larger gallery sizes aim to evaluate XBT in a more realistic and challenging retrieval scenario. We will clarify this more in the final version of the paper.

---

> > ### Author Response · Authors · 2024-11-25
> >
> > Dear reviewer nZUG,
> >
> > We appreciate your valuable comments on our paper. We have prepared a rebuttal with an updated manuscript and tried our best to address your concerns. We notice that the author-reviewer discussion period is coming to an end, and we are willing to answer any unresolved or further questions that you may have regarding our rebuttal if time is allowed.
> >
> > If our rebuttal has addressed your concerns, we would appreciate it if you would let us know your final thoughts. Additionally, we will be happy to answer any further questions regarding the paper. Thank you for your time and consideration.

---

> > > ### Comment · Reviewer_nZUG · 2024-11-26
> > >
> > > I cannot see any revisions, and my concerns still remain. I would like to keep my score.

---

> > > > ### Author Response · Authors · 2024-11-26
> > > >
> > > > Hi reviewer nZUG:
> > > >
> > > > Thank you for your time.
> > > >
> > > > Sorry about this, we just uploaded a revision that has all the changes you asked for in w1. For w4, we have been clear at the paragraph at L349 as well as L431. I thought we uploaded it but it seems it did not - our sincere apologies.
> > > >
> > > > On the remaining concerns you have, is it possible to shed more light how our answers to the weaknesses did not address your concerns. We will try our best to address the concerns if you can clarify them for us.
> > > >
> > > > Thanks.

---

### Meta-Review · Area_Chair_fTRy · 2024-12-21

**Metareview:**

The paper presents an approach to address the challenge of model upgrades in cross-modal retrieval systems by introducing Cross-modal Backward-compatible Training (XBT). This approach aims to align the embeddings of new Vision-Language Pretraining (VLP) models with those of the old models, thereby reducing the need for costly re-embedding of data, known as backfilling.

Strengths:
+ Comprehensive Experimentation: The paper demonstrates strong empirical validation through extensive experiments across multiple models and datasets.
+ Novel Perspective: The authors present an innovative approach by proposing that aligning a single modality can potentially lead to cross-modal alignment.
+ Practical Relevance: The work addresses a real-world challenge in deploying updated multimodal systems while maintaining compatibility with existing infrastructure, making it valuable for practical applications.

Weaknesses:
+ The presentation needs to be further improved, some tables or figures are not clear. The motivation for the text-only pret-training stage needs to be highlighted.
+ More ablation studies are expected. The computational analysis is limited to a comparison of training times and memory loads. A more comprehensive analysis are needed, including the trade-offs between accuracy and computational resources.
+ The experimental results are unconvincing.

**Additional Comments On Reviewer Discussion:**

After the rebuttal, the submission received mixed reviews (5566). Several reviewers didn't respond to the rebuttal and insisted on their initial ratings. Reviewer nZUG thought his/her concerns remain. Overall, there are still several aspects that can be further improved. I recommend Reject.

---

### Decision · Program_Chairs · 2025-01-22

Reject